# Associations between Workers’ Health and Working Conditions: Would the Physical and Mental Health of Nonregular Employees Improve If Their Income Was Adjusted?

**DOI:** 10.3390/medicines9070040

**Published:** 2022-07-14

**Authors:** Mariko Nishikitani, Mutsuhiro Nakao, Mariko Inoue, Shinobu Tsurugano, Eiji Yano

**Affiliations:** 1Medical Information Center, Kyushu University Hospital, Fukuoka 812-8582, Japan; 2Department of Psychosomatic Medicine, International University of Health and Welfare, Chiba 286-8529, Japan; m-nakao@iuhw.ac.jp; 3Graduate School of Public Health, Teikyo University, Tokyo 173-8605, Japan; inoue-ph@med.teikyo-u.ac.jp (M.I.); eyano@med.teikyo-u.ac.jp (E.Y.); 4Center for Health Sciences and Counseling, Kyushu University, Fukuoka 819-0395, Japan; tsurugano@chc.kyushu-u.ac.jp

**Keywords:** workers’ health, nonregular employment, self-rated health, mental health, income, socioeconomic status

## Abstract

Precarious employment can negatively affect health, but workers may be healthy if they earn enough income. This study uses equivalent disposable income and examines the interaction between income classes and employment types to clarify whether workers’ health improves as the income classes rise. In Japan, nonstandard workers, called nonregular employees, have remained high since 2013. Therefore, using data from the national cross-sectional Comprehensive Survey of Living Conditions 2013, an official survey performed in Japan, we targeted a sample of employees aged 18 to 45 who graduated during the economic recession. Our final sample included 8282 employees (4444 males and 3838 females). The health (general and mental) status indicators used the dichotomized self-rated health and scores of the K6 questionnaires scored in the national survey. The association between income and health was almost proportional. Female workers tended to improve their health as their income class increased; however, this tendency was not observed in male workers, especially nonregular employees. Although the associations were weakened by added income information on the regression models, nonregular employees always showed inferior health to regular employees. The health status of nonregular employees, especially female nonregular employees, is statistically significantly lower than that of regular employees, even when the economic class is similar. In conclusion, improving low incomes for nonregular employees could improve health challenges, but income alone may not result in the same health status for regular and nonregular employees.

## 1. Introduction

Recently, nonregular employment has become common in Japan, representing 37% of all employment [1]. It is more prevalent among “service workers” and “carrying cleaning, packaging, and related workers” than among regular workers in the same occupations in Japan. In industrial categories, larger numbers of nonregular employees are reported to be working in service-related sectors, such as “accommodations, easing, and drinking services” and “living-related and personal services and amusement services”, than regular employees [2]. The working conditions are very similar to “nonstandard employment”, which the International Labor Organization described in three employment types [3]: temporary employment, part-time and on-call work, and multiparty employment relationships. The specific features of Japanese nonregular employment create several issues concerning welfare [4,5,6], health [7,8,9], and fostering future generations [6]; the welfare and income of nonregular workers differ significantly from those of regular workers [10]. For example, full-time regular workers earned an average of JPY 2599 (almost USD 22) per hour; full-time nonregular employees earned only JPY 1269 (USD 11) in 2021 [11]. This considerable economic disparity has persisted for over a decade, as reported by the Organisation for Economic Co-operation and Development (OECD) [12]. To make matters worse, nonregular employees do not receive retirement allowances; those who work less than a certain number of hours must take out national health insurance and pension plans with (self-paid) premiums that are almost twice those of regular employment insurance [10,13]. Such systematic inferior treatment has created a socioeconomic subclass with significant health implications, where social exclusion compromises overall wellness and life satisfaction [14,15]. According to social epidemiologists, high socioeconomic status, as represented by occupation, educational background, and social class, is associated with longevity and persistent good health [16,17,18].

According to statistics, nonregular workers have increased since 1995 and have remained high, especially among women [1]. With the increasing numbers and growing concern about nonstandard workers’ inferior working conditions, many recent studies have assessed worker health. These studies show that poor socioeconomic status is associated with precarious employment and that inferior working conditions negatively affect health [4,5,7,8,9,19,20,21,22,23]. One review found that nonstandard employees are more likely to be injured at work, have higher mortality, and suffer more from mental health issues than regular employees [24]. Moreover, the Japanese population’s overall health has deteriorated as the proportion of nonregular employees has risen [25].

While the extant literature has examined nonregular employment, not all studies have explored gender differences. It is thought that female nonregular workers in Japan differ in working conditions and socioeconomic status from male nonregular workers and nonstandard female employees in other developed countries [26]. In Japan, spouses partnered with regular employees as a legally married couple often adjust spouses’ annual income to less than JPY 1,030,000, or 1,300,000 to benefit from tax exemptions and social security support [27]. Many couples have adjusted their wives’ income to take this benefit, and these generous protections allow nonregular female workers and nonworking homemakers to enjoy better health than regularly employed females [27]. Therefore, considering gender differences is essential when assessing the health of regular and nonregular employees.

In general, nonregular employees are more likely to be female, and the age distribution skews toward the younger and older ends of the spectrum [1,2,3]. Even worse, nonregular employment commonly involves changes in jobs and workplaces, varying work times, relatively low work hours and days, and a lack of human resources management [3]. Because of the precarious status of nonregular employees, their work conditions are obscure, even to employers, researchers, and society. Moreover, their health and economic situations have long been ignored because, usually, their low incomes are thought to be a result of their shorter working hours. Thus, the associations between income considering work time and health have not been comprehensively assessed, particularly among nonregular employees.

We explored the associations between workers’ health and economic status as their working conditions, considering employment and life statuses according to gender. Although nonregular employees suffer from low incomes, regular employees usually work long hours [28,29,30], and working hours should be considered. Therefore, to identify the association between health and economic status, we hypothesized that if economic status were identical after working time was adjusted together with other sociodemographic factors, the health status would be comparable, and we conducted multivariable regression analysis.

This study shows the results that individual economic benefits partly alleviate workers’ health problems, as well as evidence that economic benefits alone cannot solve health problems caused by workers’ employment status disparities. In this context, this paper provides interesting avenues for future research assessed by both medicine and society.

## 2. Experimental Section

This was an observational study using national survey data. All data were coded and anonymized before release; therefore, no ethics review was required. We targeted data in 2013 when the number of nonregular employees remained high; we processed the application for nonpurpose and obtained and analyzed individual data.

### 2.1. Data Source and Samples

This study used data from a national cross-sectional survey, the Comprehensive Survey of Living Conditions, an official survey of the Japanese population [31]. The survey has been conducted annually since 1986 to obtain basic data on household structures. A large-scale survey is performed every 3 years to assess living conditions, healthcare status, access to and use of medical services, welfare, and pensions. The data help to develop health, welfare, and labor policies. All participants are members of households selected via cluster sampling from districts subjected to a population census in the previous census year [31].

In 2013, household and health questionnaire forms were distributed to almost 295,367 households and were returned by 235,012 (79.6%) [31]. Income questionnaires were distributed to randomly selected households from various districts. The sampling percentage was approximately 12%, and the survey was completed by 26,387 households, representing a 72.5% response rate. We obtained permission to use individual-level data from the large-scale 2013 survey for purposes other than those initially intended by Japan’s Ministry of Health, Labour, and Welfare. We used data from all three forms.

In the Japanese labor market, the mainstream system is intensive graduate recruitment. Therefore, the economic state strongly influences whether or not young graduates can work as standard employment. If students graduate during a recession, many start their careers as nonregular employees. Due to the intensive recruitment of graduates, these nonregular employees have a rare chance to become regular employees in their later careers. This study used the ratio of job offers to job seekers as an index of recession. People who graduated from master’s courses in 1992 and began working when the ratio of job offers to applicants was <1 were 45 years old in 2013. The ratio of job offers to job seekers was usually below 1 until it reached 1.11 in 2014, except that it was 1.06 in 2006 and 1.02 in 2007. Therefore, from the 2013 dataset, we targeted employees aged 18–45, including those who graduated from high school at 18.

Workers who did not enroll in their own health insurance schemes were excluded in this study, as they were usually dependent as family members of a regular employee and received social security benefits and tax exemptions owing to their low income. To avail of these economic benefits, they intentionally save their working hours and income [26,27]. The association between economic social class and health could be clearly assessed by excluding the fake artifact. Therefore, the study included only independent workers who worked without unnecessarily adjusting their working hours.

For employment status, survey participants were defined by their engagement in “income-earning jobs” (salaried workers) and by an employment status of “employee.” Regarding financial information, we included only survey participants who returned the income questionnaire.

Among target subjects, we excluded participants who did not have information on employment status and working hours, those who did not answer health status questions, and those who had incomplete surveys. The study participants totaled 111,956 employees (59,882 males and 52,074 females). After including income information, the final group comprised 8282 employees (4444 males and 3838 females). When exploring health status, males and females were evaluated separately because of biological and sociological differences.

### 2.2. Variables

#### 2.2.1. Employment Groups

All subjects were classified as either regular or nonregular employees, including part-time work (being paid hourly, dispatch work, being on a temporary payroll with a contract for less than 1 year, or being hired by the day). Regular employment was defined as any regular work contract, typically associated with an annual or monthly salary to retirement age (65–70 years), when a pension would commence.

#### 2.2.2. Health Status

General and mental health statuses were used as overall health indicators in the Comprehensive Survey of Living Conditions questionnaire. General health was assessed using self-rated health (SRH) questionnaires that explored general health by asking, “What is your current health status?” We classified the general health status as either worse (1) than “rather poor” or better (0) than “fair”, based on the responses on a five-point Likert scale: “excellent”, “very good”, “fair”, “rather poor”, or “poor.” SRH is a well-validated predictor of objective general health outcomes from mortality to health-related behaviors [32,33] and has been used to examine the link between health and job insecurity [7,14,25,33,34]. Mental health status was inferred from answers to the Japanese version of the K6 questionnaire [8,9,35], which detects types of psychological distress, such as mood disorders and depressive symptoms. The value was calculated by summing the scores from answers to six questions regarding experiences during the past 30 days; relevant keywords included “nervous”, “desperate”, and “awkward”, and subjects gave self-scores from 0 (never) to 4 (all the time).

#### 2.2.3. Economic Status

This study estimated and compared the relationship between social income class and health status for each employment type. As the indicator of the social income classes, equivalent disposable income from all households (26,387 households) was used and divided into quartiles. The equivalent disposable income was the annual after-tax household income divided by the square root of the number of household members. Because the distribution was curvilinear, these income data were converted into dummy values from 1 to 4 based on the quartile order. The geometric means and 95% confidence intervals of each quartile category were USD 963 (942–985), 1984 (1973–1994), 2919 (2906–2931), and 4629 (4587–4671) by a conversion rate of USD 1/JPY 100 in 2013. The dummy numbers of income class were entered into a multivariate regression model as explanatory variables.

#### 2.2.4. Sociodemographic

In addition to employment status and equivalent disposable income, we evaluated other factors that might affect health and work, including gender, individual income, work hours, and life-related factors, such as age, spousal status, children under 20 years old living at home, and living status (alone or not).

### 2.3. Data Analysis

Basic characteristics and health status by gender were compared between regular and nonregular employees. The chi-square test was employed to compare differences in categorical variables, and we used Wilcoxon’s rank-sum test to compare differences in continuous variables.

Next, we performed a multivariate regression analysis of the association between economic and health statuses. For the statistical estimation of the effects of economic status on health status, we used logistic regression for general health status and the dependent variable with a binary value of poor (1) or fair and better (0). Ordinary least squares linear regression was used for mental health, the continuous dependent variables of K6 scores. The predicted margin scores in health outcomes used the ad hoc function of the Stata software designed for regression analysis (ver. 16.0; StataCorp, College Station, TX, USA). To exclude the sampling bias and benefit from the large numbers of survey participants, we first conducted a regression model without income information (Model 1), followed by another model with income information (Model 2), to predict two health indicators (SRH and K6) for each gender group. Both regression models included confounding factors: working hours, age, spousal status, whether children under 20 years old live at home, and living alone (dichotomous values were employed for the latter three variables). To estimate the impact of employment status (regular or nonregular) on health status considering economic status, we entered interaction terms of employment and economic statuses into the model. All data analyses were conducted using the Stata software, and a two-tailed *p*-value < 0.05 was taken to indicate significance.

## 3. Results

Most basic characteristics differed significantly between regular and nonregular employees, regardless of gender, those including both with and without income information (the upper sections of Table 1 and Table 2), and those with income information (the lower sections of Table 1 and Table 2). However, most features differed between male and female employees. More male nonregular employees were younger than regular employees, lived alone, were not married, and had no children under 20 years old (Table 1). Nonregular female employees were older than regular female employees; they also lived with family more often, were married, and had children (Table 2).

Nonregular employees of both genders had lower incomes and fewer work hours than regular employees. Regarding health indicators, nonregular employees exhibited poorer general health and more mental health problems than regular employees. In this simple comparison among participants with income information, although the number of analyzed participants was decreased, the mental health status still differed significantly between male regular and nonregular employees (the lower part in Table 1).

Although we divided employees according to the same definitions of income quartiles, there were significant differences in their numbers and percentages within each quartile between regular and nonregular employees (both males and females; *p* < 0.001, chi-square test; Table 1 and Table 2).

Model 1 of the regression analysis, adjusted for confounding factors other than income, showed significant associations between health indicators and employment status for males and females, indicating significantly poorer general and mental health for nonregular employees.

After adding income information, health status predicted by regression analysis after adjustment for confounding variables (Model 2 in Table 3 and Table 4; Figure 1 and Figure 2) indicated that both health indicators were no longer associated with employment status in male employees. Instead, the health status of employees in the higher-income classes was not good unexpectedly compared with the lowest-income ranks. Predicting and plotting the interaction between employment status and income from a regression model (Figure 1) showed that nonregular employees were not statistically significant in higher-income groups; however, there was a tendency toward poor general health (Model 2 in Table 3; Figure 1). Regarding mental health, the health indicators for nonregular employees also tended to be poor with the higher-income class, although it was not a significant interaction.

The association between general health and employment status disappeared with a multivariate analysis of female nonregular employees with income information. However, even when income information was added, nonregular female employees had inferior mental health (K6 has a high score) than regular employees. Moreover, nonregular female employees experienced poor general health compared with regular employees in upper-income classes of the second quartile (Model 2 in Table 4; the left in Figure 2). Nonregular female employees in the upper classes experienced significantly better mental health (lower score in K6) than females in the first lower-income classes (Figure 2, right). There was no similar difference in female regular employees, and this was the significant effect of the interaction between employment status and income (Figure 2, right).

## 4. Discussion

Generally, the association between income and health is expected to be simply proportional; the higher an individual’s income class, the better his or her health. These tendencies were evident in female workers younger than 45 years old, reflected in the general health status based on SRH. There was no similar statistically significant association in female employees regarding mental health status, but it was indicated partly among nonregular employees. Conversely, both health statuses of male workers did not indicate a similar association with income classes. Instead, according to upper-income classes, male nonregular employees tended to have poorer health than the lower-income class, though the disparity was not statistically significant.

It is reasonable to expect that an employee from a higher economic status would enjoy better health and vice versa. Our observations on female employees younger than 45 align with previous studies: inferior socioeconomic status (low income) compromised health [9,17]. Generally, female workers spend 2.5 times more time on housework and family care than male workers [3,27]. Therefore, female employees work fewer hours than the average male. In Japan, because females cannot have enough time to work outside of the home, their average income is significantly lower than males, reflecting the significant economic disparity, which has long been suggested by the OECD [3,36]. Moreover, nonregular employees have the lowest incomes among women [36]. Therefore, female employees may be more likely to improve their economic life and health if they escape the sacrifice of income inequality in society and raise their income classes.

Our observation that the highest income group had poorer health than the next-highest group is at first controversial. However, ecologically, the relationship between income and health indicators, such as gross domestic product and life expectancy, would potentially become saturated at higher economic levels [37]. For example, the longer a worker’s career, the higher the income; however, as workers grow old, their health is potentially affected by age-related issues. Additionally, workers might have to work long hours to obtain a high income. Predicted health scores were also adjusted for other confounding variables, including age and work hours. Therefore, workers in the highest-income group may experience burdens other than long working hours and other individual sociodemographic factors, such as older age, high levels of family responsibility, and resident status, like living alone. Nonetheless, the hypothetical association between males’ health and economic indicators may have been less due to their relatively higher incomes than women. In addition, there may be a male-specific income–health link. Unlike women, regular male employees are older than nonregular employees, live with their families, and work longer hours (Table 1); therefore, higher-income classes might have to generate household income for their other family members. It may be necessary and time consuming, which can become mentally burdensome. Therefore, the health indicators could potentially not improve with a high-income class.

In contrast, workers who belong to the higher-income class in an unstable state of nonregular employment with no spouse or children, have a weak family relationship, and are living alone (Table 1) may face health effects due to social isolation and loneliness. The problem of isolation for Japanese men is famous, mainly for the elderly who preceded their spouses; however, even younger unmarried individuals can face the same isolation problem. Furthermore, in precarious employment, there may be pressure to earn as high an income as a regular employee, and even if the income is high, the health condition may be poor.

This study showed that adjusting income weakens the link between employment patterns and health and reduces the negative health effects of adverse employment patterns, such as nonstandard employment. However, the mental health of female nonregular employees was still worse than that of regular employees, even after adjusting their income. Earlier reports found that regular employees were generally healthier than those not regularly employed [4,5,7,8,9,19,20,21,22,23]. A possible explanation is that nonregular female employees may be burdened by work and family [27,38,39]. Females in demanding jobs are at higher psychological risk than others because of an effort–reward imbalance [40] caused by comparatively low income. Worldwide, female stressors include work and family [39,41,42,43]; female employees may bear more responsibility and feel guiltier when balancing work and family than nonregular female employees, compromising their mental health.

Centering on female mental health, nonregular employees exhibited poorer health than regular employees, as has been previously reported [4,5,7,8,9,19,20,21,22,23], reflecting a clear socioeconomic disparity (OECD). In terms of work time, regular male Japanese employees work notoriously long hours [29,30], and it is simultaneously understood that Japanese female workers balance their time between home chores and working outside. Therefore, we carefully considered both work hours and family background when assessing the health status, and we performed stratification and adjustment before regression; however, nonregular employees still exhibited poorer health. Recently, the Japanese government has implemented a “Work Style Reform” program through the Ministry of Health, Labour, and Welfare (2018) to improve the working conditions of both regular and nonregular employees. For example, they recommend that wages not depend on the type of employment; however, this has not been fully effective, and the health of nonregular employees remains poor, perhaps due to precarious situations.

Our work had several limitations. First, we lacked education information. Academic backgrounds should be considered a confounding factor when assessing relationships between health status, employment, and income; higher-paying jobs relate to academic careers, and several lifestyle features and chronic diseases are affected by educational status [17,19,20,23]. In addition, the dominant concept of health status, like quality of life and life satisfaction, could better indicate the relationship between income status and working time. Although we aimed to adjust working conditions along these lines, our efforts may have been inadequate; thus, future work should consider educational background and quality of life.

Another limitation is that all data were self-reported. Objective measures, especially health parameters, are desirable. Although it has been assessed subjectively, in a recent study, SRH has shown a significant association with blood and urine biomarkers, indicating inflammation, abnormal metabolism, lifestyle-related and environmental exposure, and chronic diseases [44]. Thus, SRH used in this study might be an effective predictive measure of human health status. In terms of working conditions, objective measures may be available from human resources departments; however, some employees, such as dispatch workers, are not managed by such departments because their salaries come from procurement rather than personnel budgets. Moreover, work time records vary by employment type and company, and it is challenging to record discretionary workers’ hours. Furthermore, workers and employers do not always record unpaid overtime (also termed service overtime). Therefore, working conditions were subjective, and our data should be interpreted with caution.

Finally, this was a cross-sectional study using existing survey data; thus, we could not identify causal relationships between health and working conditions. Again, more longitudinal cohort studies are required because we could not identify causal relationships.

Additionally, this study’s dataset might appear outdated because we analyzed the national dataset collected in 2013. The social situation surrounding workers might have changed since 2013, and the interpretation of the findings may need attention for such social changes. However, the rate of nonregular workers across the entire labor force has been steady at 37% since 2013, and the average income has also drifted below USD 45,000 at a similar level in Japan. Therefore, we believe that the findings still apply to the current workers’ employment and economic status.

## 5. Conclusions

We found associations between health and income; healthy workers had better economic status than unhealthy workers. Regarding female workers, especially nonregular employees, health was associated with income classes. Moreover, although their economic status was considered, female nonregular employees had poorer health than regular employees. Additionally, nonregular male employees tended to be less healthy even if they belonged to a relatively higher-income class. Overall, the health of nonregular employees, regardless of gender, always looks worse than that of regular employees if income conditions are not considered. Such gaps should be minimized in the future by providing the same payment for the same jobs, regardless of differences in gender and employment status.

## Figures and Tables

**Figure 1 medicines-09-00040-f001:**
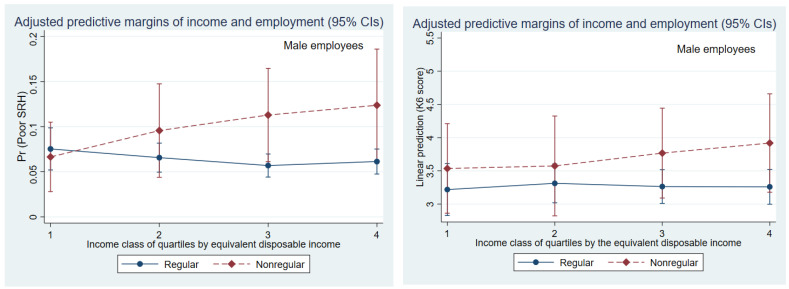
Association between economic status (equivalent disposable income quartile) and poor health (predicted self-rated health and K6 scores) by employment status in male workers.

**Figure 2 medicines-09-00040-f002:**
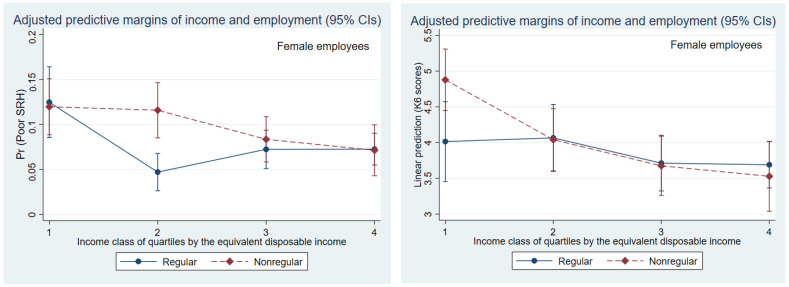
Association between economic status (equivalent disposable income quartile) and poor health (predicted self-rated health and K6 scores) by employment status in female workers.

**Table 1 medicines-09-00040-t001:** Basic characteristics of male employees (upper: all survey participants with and without income information; lower: participants with income information).

All Survey Participants: Independent Employees Aged between 18 and 45 in 2013	Regular Employees (*n* = 51,053)		Nonregular Employees (*n* = 8829)		*p* *
Age: median, 25–75% (years)	36 (30–41)		29 (23–36)		<0.0001
Living alone (yes)	6054	(12%)	1459	(17%)	<0.001
Spouse (yes)	30,296	(59%)	1515	(17%)	<0.001
Children < 20 years of age (yes)	26,961	(53%)	2718	(31%)	<0.001
Working hours: median, 25–75% (h/week)	48 (40–55)		40 (24–46)		<0.0001
General health (“poor” or “rather poor” by self-rated health)	3458	(7%)	615	(7%)	0.507
Mental health (K6): median, 25–75% (score ^a^)	1 (0–5)		2 (0–6)		<0.0001
Sampled participants to answer economic status among all	(*n* = 3805)		(*n* = 639)		
Individual income ^b^ (×USD 1000/year): geometric mean (95% CI ^c^)	38.82 (38.00, 39.66)		15.58 (14.62, 16.61)		<0.0001
Equivalent disposable income ^b^ (×USD 1000/year): geometric mean (95% CI ^c^)	26.55 (26.11, 27.00)		21.63 (20.55, 22.77)		<0.0001
Income class of 1st quartile in all households’ equivalent disposable income	505	(13%)	175	(27%)	<0.001
2nd quartile	918	(24%)	139	(22%)	
3rd quartile	1224	(32%)	176	(28%)	
4th quartile	1158	(30%)	149	(23%)	
Age: median, 25–75% (years)	36 (30–41)		30 (24–38)		<0.0001
Living alone (yes)	305	(8%)	84	(13%)	<0.001
Spouse (yes)	2280	(60%)	109	(17%)	<0.001
Children < 20 years of age (yes)	2017	(53%)	177	(28%)	<0.001
Working hours: median, 25–75% (h/week)	48 (40–55)		40 (30–47)		<0.0001
General health (“poor” or “rather poor” by self-rated health)	244	(6%)	54	(8%)	0.1733
Mental health (K6): median, 25–75% (score ^a^)	1 (0–5)		2 (0–6)		<0.0001

* Wilcoxon’s rank-sum test was used to compare continuous variables, such as age, working hours, mental health score, and income, and the chi-square test was used to compare categorical variables between regular and nonregular employees. ^a^ Total K6 score from 0 = all “no” responses to 24 = all “yes” responses. ^b^ The income exchange rate from Japanese yen to US dollars is based on the average conversion rate of USD 1/JPY 100 in 2013. ^c^ CI: confidence interval.

**Table 2 medicines-09-00040-t002:** Basic characteristics of female employees (upper: all survey participants with and without income information; lower: participants with income information).

All Survey Participants: Independent Employees Aged between 18 and 45 in 2013	Regular Employees (*n* = 26,807)		Nonregular Employees (*n* = 25,267)		*p* *
Age: median, 25–75% (years)	33 (27–39)		36 (29–41)		<0.0001
Living alone (yes)	2826	(11%)	1337	(5%)	<0.001
Spouse (yes)	10,457	(39%)	14,378	(57%)	<0.001
Children < 20 years of age (yes)	11,379	(42%)	15,944	(63%)	<0.001
Working hours: median, 25–75% (h/week)	40 (40–48)		30 (20–40)		<0.0001
General health (“poor” or “rather poor” by self-rated health)	2196	(8%)	2319	(9%)	<0.001
Mental health (K6): median, 25–75% (score ^a^)	2 (0–6)		2 (0–6)		<0.0001
Sampled participants to answer economic status among all	(*n* = 2051)		(*n* = 1787)		
Individual income ^b^ (×USD 1000/year): geometric mean (95% CI ^c^)	26.36 (25.58–27.17)		12.25 (11.82–12.68)		<0.0001
Equivalent disposable income ^b^ (×USD 1000/year): geometric mean (95% CI ^c^)	28.63 (27.90–29.37)		21.1 (20.61–21.83)		<0.0001
Income class of 1st quartile in all households’ equivalent disposable income	271	(13%)	477	(27%)	<0.001
2nd quartile	383	(19%)	454	(25%)	
3rd quartile	558	(27%)	501	(28%)	
4th quartile	839	(41%)	355	(20%)	
Age: median, 25–75% (years)	33 (27–39)		36 (29–41)		<0.0001
Living alone (yes)	202	(10%)	70	(4%)	<0.001
Spouse (yes)	721	(35%)	890	(50%)	<0.001
Children < 20 years of age (yes)	813	(40%)	1062	(59%)	<0.001
Working hours: median, 25–75% (h/week)	41 (40–48)		32 (24–40)		<0.0001
General health (“poor” or “rather poor” by self-rated health)	160	(8%)	168	(9%)	0.077
Mental health (K6): median, 25–75% (score ^a^)	2 (0–6)		2 (0–6)		0.8375

* Wilcoxon’s rank-sum test was used to compare continuous variables, such as age, working hours, mental health score, and income, and the chi-square test was used to compare categorical variables between regular and nonregular employees. ^a^ Total K6 score from 0 = all “no” responses to 24 = all “yes” responses. ^b^ The income exchange rate from Japanese yen to US dollar based on the average conversion rate of USD 1/JPY 100 in 2013. ^c^ CI: confidence interval.

**Table 3 medicines-09-00040-t003:** Effect of employment, working conditions, and demographic characteristics on health indicators in the regression models for prediction among male workers.

	Poor General Health: Odds Ratio (95% CI ^a^)		Mental Health (K6): Coefficient (95% CI)	
	Model 1 (*n* = 59,882)	Model 2 (*n* = 4444)	Model 1 (*n* = 59,882)	Model 2 (*n* = 4444)
Nonregular employees	1.242 * (1.126, 1.369)	0.873 (0.430, 1.775)	0.540 * (0.432, 0.651)	0.318 (−0.463, 1.098)
Income class of 1st quartile in all households	-	1.000 (reference)	-	0.000 (reference)
2nd quartile	-	0.862 (0.563, 1.320)	-	0.094 (−0.393, 0.580)
3rd quartile	-	0.740 (0.490, 1.117)	-	0.045 (−0.420, 0.510)
4th quartile	-	0.801 (0.528, 1.216)	-	0.041 (−0.432, 0.514)
Interaction term (employment × income class)	-	1.000 (reference)	-	0.000 (reference)
Nonregular × 2nd quartile	-	1.727 (0.664, 4.493)	-	−0.056 (−1.166, 1.055)
Nonregular × 3rd quartile	-	2.424 (0.986, 5.959)	-	0.186 (−0.864, 1.236)
Nonregular × 4th quartile	-	2.485 (0.972, 6.353)	-	0.342 (−0.753, 1.437)
Age	1.026 * (1.021, 1.032)	1.035 * (1.016, 1.055)	0.012 * (0.006, 0.017)	0.004 (−0.017, 0.025)
Living alone	1.369 * (1.240, 1.512)	1.447 (0.951, 2.202)	1.244 * (1.129, 1.359)	0.964 * (0.467, 1.460)
Spouse	1.063 * (0.964, 1.172)	1.174 (0.807, 1.707)	0.005 (−0.101, 0.111)	0.071 (−0.334, 0.475)
Children < 20 years of age	0.965 (0.884, 1.053)	0.988 (0.703, 1.388)	−0.215 * (−0.309, −0.121)	−0.117 (−0.483, 0.250)
Working hours	1.005 * (1.003, 1.007)	1.000 (0.991, 1.010)	0.007 * (0.004, 0.010)	0.002 (−0.009, 0.012)

^a^ CI: Confidence interval. * *p* < 0.05.

**Table 4 medicines-09-00040-t004:** Effect of employment, working conditions, and demographic characters on health indicators in the regression models for prediction among female workers.

	Poor General Health: Odds Ratio (95% CI ^a^)		Mental Health (K6): Coefficient (95% CI)	
	Model 1 (*n* = 52,074)	Model 2 (*n* = 3838)	Model 1 (*n* = 52,074)	Model 2 (*n* = 3838)
Nonregular employees	1.098 * (1.024, 1.177)	0.954 (0.597, 1.523)	0.397 * (0.308, 0.487)	0.863 * (0.159, 1.566)
Income class of 1st quartile in all households	-	1.000 (reference)	-	0.000 (reference)
2nd quartile	-	0.344 * (0.192, 0.618)	-	0.051 (−0.670, 0.772)
3rd quartile	-	0.546 * (0.338, 0.881)	-	−0.302 (−0.977, 0.373)
4th quartile	-	0.547 * (0.350, 0.855)	-	−0.324 (−0.967, 0.318)
Interaction term (employment × income class)	-	1.000 (reference)	-	0.000 (reference)
Nonregular × 2nd quartile	-	2.797 * (1.369, 5.715)	-	−0.888 (−1.824, 0.048)
Nonregular × 3rd quartile	-	1.225 (0.645, 2.329)	-	−0.901 * (−1.789, −0.013)
Nonregular × 4th quartile	-	1.029 (0.525, 2.018)	-	−1.024 * (−1.923, −0.125)
Age	1.029 * (1.024, 1.034)	1.031 * (1.013, 1.049)	0.013 * (0.007, 0.019)	−0.012 (−0.035, 0.010)
Living alone	1.250 * (1.115, 1.402)	1.145 (0.730, 1.795)	1.038 * (0.886, 1.190)	1.275 * (0.674, 1.877)
Spouse	0.939 (0.866, 1.017)	0.985 (0.729, 1.331)	−0.329 * (−0.432, −0.226)	−0.010 (−0.405, 0.385)
Children < 20 years of age	1.005 (0.932, 1.084)	1.032 (0.775, 1.374)	−0.071 (−0.166, 0.024)	−0.008 (−0.372, 0.357)
Working hours	0.999 (0.997, 1.002)	1.014 * (1.004, 1.025)	0.006 * (0.002, 0.009)	0.001 (−0.012, 0.014)

^a^ CI: Confidence interval. * *p* < 0.05.

## Data Availability

Not applicable.

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
