# Peer review of "Associations between Workers’ Health and Working Conditions: Would the Physical and Mental Health of Nonregular Employees Improve If Their Income Was Adjusted?"

_medicines, 2022, doi:10.3390/medicines9070040_

Round 1

Reviewer 1 Report

Dear authors - Your study is interesting, with a well-written and clearly organized manuscript! I have three minor comments for your consideration. First, please consider adding a sentence or two in the introduction providing examples of industries or occupations that typically provide "non-regular employment" in Japan (e.g., agriculture, transportation, retail, etc.). Secondly, you note that the national cross-sectional survey is conducted every three years and that data are from 2013. Why not use more recent data? Please consider noting that the data being collected almost a decade ago creates a minor limitation. Thirdly, please consider editing Tables 1 and 2 to more clearly explain the upper and lower sections. In lines 237 -240, you note "In this simple comparison among participants with income information, the mental health status  differed significantly between male regular and non-regular employees (the lower part in Table 1). " The tables have repeated variables (i.e., Age; Living alone; Spouse; Children <20 years; Working hours; General Health; Mental Health) because the upper portion is from participants with income information and the lower portion is without income information? Please clarify with the tables themselves and the text. 

Reviewer 2 Report

The authors presented a cross-sectional study involving data by a large sample of Japanese workers, regular and non-regular employees.

The aim of their research was to explore the relationships between workers’ health and working conditions.

Results provided evidence on better health in workers with better economic status, as expected by the authors.

The study provides further knowledge to the detrimental effects on workers’ mental health and wellbeing of low incomes.

The introductory part is well written and grounded and the methodological section is sound.

I only suggest a minor revision.

Lines 95-96: limits should be all described in the final part.

Author Response

This manuscript is a resubmission of an earlier submission. The following is a list of the peer review reports and author responses from that submission.